# EXTRACTING ROBUST ON-MANIFOLD INTERACTIONS ENCODED BY NEURAL NETWORKS

## ABSTRACT

This paper aims to extract faithful interactions between input variables encoded by a deep neural network (DNN). Recent studies (Ren et al., 2023d; Li & Zhang, 2023b) provided lots of mathematical evidence to support that interactions can be roughly considered as primitive inference patterns encoded by a DNN, given that a small number of interactions can accurately explain the network outputs on any randomly masked samples. However, the instability of interactions to small perturbations on the input still hinders people from taking interactions as rigorous primitives for the network inference. Therefore, in this paper, we propose to extract on-manifold interactions, which are shared by different perturbed inputs. The extracted on-manifold interactions can also explain primitives for adversarial vulnerability. The code will be released after the acceptance of the paper.

## 1 INTRODUCTION

In recent years, explainable artificial intelligence (XAI) has been an issue of great interest. Within the realm of XAI, one of the ultimate questions is whether the inference logic used by a well-trained deep neural network (DNN) can be faithfully explained by a set of symbolic concepts or primitives (Zhou et al., 2016; Wu et al., 2018; Kim et al., 2018). This question involves both mathematics and cognitive science. Up to now, people have not reached a consensus on the definition of concepts in DNNs, due to the inherent challenge of formulating concepts in human cognition.

In spite of that, if we ignore cognitive issues, some recent research in interactions has provided an alternative way of analyzing the knowledge or primitives encoded by DNNs. Ren et al. (2023a) defined the measurement of interactions between input variables encoded by a DNN. For example, let us consider the sentence "*she always sees the world through rose-colored glasses*" as the input of a trained DNN, where each word is considered as an input variable. In this sentence, the DNN may encode the co-occurrence of words in $S = \{through, rose, -colored, glasses\}$ to represent the meaning of being optimistic. Thus, $S$ is considered to contain an AND interaction between these four words, which has a numerical effect (denoted by $I(S)$) to push the output towards optimism.

Although there is no justification for whether such interactions fully align with human cognition, these studies have nonetheless provided various evidence for considering these interactions as primitives that are used by the DNN to compute the output score.

(1) **Sparsity.** Ren et al. (2023d) have proven that, under some common conditions[1], a well-trained DNN usually learns just a small number of interactions with salient effects. Effects of all other interactions *w.r.t.* most subsets of input variables are almost zero.

(2) **Universal matching.** Li & Zhang (2023a) have further proven that the complex inference logic (*i.e.*, the varying output scores on an exponential number of masked input samples) of the DNN can be universally mimicked/matched by such a small number of interactions.

(3) **Generalization power.** Li & Zhang (2023b) have demonstrated that these interactions have good generalization power and discrimination power.

---

[1]There are three assumptions in (Ren et al., 2023d): (1) The high-order derivatives of the model output *w.r.t.* the input are all zero; (2) Let us randomly mask some input variables to generate lots of masked samples. The average confidence of the model inference over different masked samples monotonically decreases when we mask more input variables. (3) The decreasing speed of the average inference confidence is not faster than a polynomial function along with the ratio of input variables being masked.

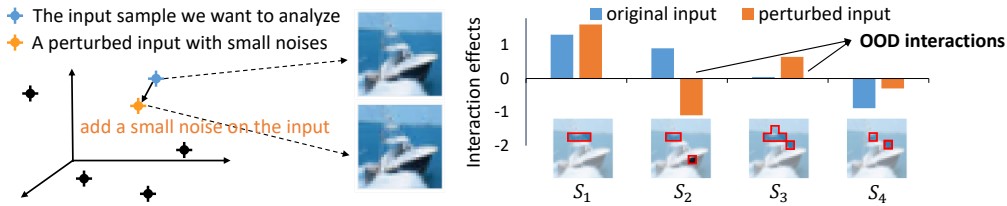

Figure 1: Interactions extracted from (Li & Zhang, 2023a) are sensitive to noises in the input. Small noises make the method extract out-of-distribution (OOD) interactions. Red rectangles on the right show some sets $(S_1, \ldots, S_4)$ of image patches that have different interactions in the noisy input.

The above proofs and experimental findings have partially guaranteed the worthiness of interactions. Please see the related work section for how interactions are used to explain deep learning.

However, the previously defined interaction is usually not stable to small perturbations on the input, which hinders the interactions from being fully considered as primitives encoded by a DNN. Specifically, given a DNN and an input sample (the blue dot in Figure 1), we find that small perturbations on the input (the orange dot in Figure 1) may make the algorithm extract a different set of interactions, *i.e.*, generating many out-of-distribution interactions as shown in Figure 1. Although these interactions can still well match the network output following the aforementioned universal-matching property, such instability of the extracted interactions hinders people from rigorously taking interactions as primitive patterns of the network inference.

To this end, ideal interactions are supposed to be stably extracted under different variations of the input, so that we can consider such interactions as common primitives encoded by the DNN. In fact, such stable interactions can be analogic to on-manifold features in a DNN. When the input is randomly perturbed[2], features in intermediate layers of the DNN usually primarily change along some specific directions, rather than change along any arbitrary direction. This phenomenon indicates the existence of a feature manifold. Similarly, given a perturbed input, the interactions are also expected to be stably extracted from a specific manifold. Such on-manifold interactions are considered robust to perturbations. In other words, we hope to extract similar sets of salient interactions from differently perturbed inputs, rather than obtain many out-of-distribution (OOD) interactions.

Therefore, in this study, we propose a method to extract robust on-manifold interactions encoded by a DNN. We perturb each input sample with different random noises, and then extract similar sets of interactions from different perturbed inputs. To this end, we revise the objective of extracting interactions to penalize OOD interactions and meanwhile maintain the sparsity of interactions.

Furthermore, the robustness of on-manifold interactions to noises makes the on-manifold interaction more likely to explain primitives of adversarial vulnerability than traditional interactions. To be precise, we can disentangle the overall change of network output under an input perturbation into the changes of different interaction effects, *i.e.*, $\Delta v(x) = \sum_S \Delta I(S)$, where $v(x)$ denotes the network output on the input $x$. When the input $x$ is perturbed by a noise, $\Delta v(x)$ denotes the change of the network output, and $\Delta I(S|x)$ denotes the change of interaction effects of $S$. In particular, our on-manifold interactions well explain findings in previous studies (Ren et al., 2021), *i.e.*, complex interactions between lots of input variables are usually more vulnerable to adversarial attacks than simple interactions between a few input variables. The on-manifold interactions also help us understand the effectiveness of adversarial training, *i.e.*, enhancing the robustness of complex interactions.

## 2 RELATED WORKS

Extracting interactions between input variables has been an emerging direction for explaining DNNs in recent years. Some studies (Murdoch et al., 2018; Singh et al., 2019; Jin et al., 2020) quantified the numerical contribution of a subset of input variables to the output features at each layer in LSTMs. Sundararajan et al. (2020); Janizek et al. (2021); Tsai et al. (2022) defined interactions between input variables of a DNN in the framework of game theory. Recently, Ren et al. (2023a) introduced the AND interaction between input variables based on Harsanyi dividend (Harsanyi, 1959), and considered the interaction between input variables as concepts encoded in the DNN.

---

[2]Here, we do not consider adversarial perturbations.

The above interactions have been used to explain deep learning in various perspectives. Ren et al. (2023c) proved that Bayesian neural networks were less likely to encode complex interactions. Deng et al. (2022) identified a bottleneck of the DNN's capacity in learning interactions, *i.e.,* DNNs tended to encode bivariate interactions with very simple or very complex contexts, while bivariate interactions with contexts of intermediate complexity were hard to learn. Different types of interaction have been used to explain the generalization ability (Zhang et al., 2021), adversarial robustness (Ren et al., 2021), and adversarial transferability (Wang et al., 2021) of DNNs. Ren et al. (2023b) have used interactions to learn optimal baseline values for computing Shapley values (Shapley, 1953).

# 3 EXTRACTING ROBUST INTERACTIONS

## 3.1 PRELIMINARY: INTERACTIONS

Given an input $x \in \mathbb{R}^n$ and a function $v : \mathbb{R}^n \to \mathbb{R}$, the function may encode the interaction between different input variables. For example, in an input image with $n$ patches, the function may encode the interaction between different patches to represent certain objects. Li & Zhang (2023a) have defined two types of interaction effects between input variables that affect the function output $v(x)$.

**AND interaction.** We use $N = \{1, 2, \ldots, n\}$ to denote the set of the $n$ variables in $x$. Let us use AND interactions between input variables to explain the target function $v(x)$. For each subset of input variables $S \subseteq N$, the numerical effect of the AND interaction between variables in $S$ on $v(x)$ is defined as Harsanyi dividend (Harsanyi, 1959), as follows.

$$I_{\text{and}}(S|x) = \sum_{T \subseteq S} (-1)^{|S|-|T|} v(x_T) \tag{1}$$

where $|S|$ denotes the number of input variables in $S$. In particular, $I_{\text{and}}(\emptyset|x) = v(x_\emptyset)$. For each subset $T \subseteq S$, $x_T$ denotes a masked input where we mask variables in $N \setminus T$.[3] Then, $v(x_T)$ denotes the function output on the masked input $x_T$. Accordingly, $v(x_N) = v(x)$ denotes the function output on the original input $x$, and $v(x_\emptyset)$ denotes the output when we mask all variables in the input.

Intuitively, the above interaction represents the AND relationship between input variables. For example, in the sentence "*she always sees the world through rose-colored glasses,*" the co-appearance of words in $S = \{through, rose, \text{-}colored, glasses\}$ successfully activates an AND relationship, and makes an interaction effect $I_{\text{and}}(S|x)$ to push the output towards the meaning of being optimistic. Masking any word in $S$ will break the AND relationship and remove the effect, *i.e.*, making $I_{\text{and}}(S|x_{\text{masked}}) = 0$. If $I_{\text{and}}(S|x) > 0$ (or $I_{\text{and}}(S|x) < 0$), it means that the interaction in $S$ has a positive (or negative) effect on the output score $v(x)$. If $I_{\text{and}}(S|x) \approx 0$, it indicates that the AND interaction within $S$ has almost no effect on the output.

**OR interaction.** Li & Zhang (2023a) have also extended the AND interaction to define OR interactions to explain the function $v(x)$, as follows.

$$I_{\text{or}}(S|x) = -\sum_{T \subseteq S} (-1)^{|S|-|T|} v(x_{N \setminus T}) \tag{2}$$

The OR interaction represents the OR relationship among input variables. For example, to predict the sentiment of the sentence "*it is really exciting and touching*", a function may encode the OR interaction in $S = \{exciting, touching\}$ to represent the positive sentiment. The existence of either word in $S$ will make an effect $I_{\text{or}}(S|x)$ on the output score. In particular, $I_{\text{or}}(\emptyset|x) = v(x_\emptyset)$.

For each subset $S \subseteq N$ with interaction effects $I_{\text{and}}(S|x)$ and $I_{\text{or}}(S|x)$, its interaction order is defined as $\text{order}(S) = |S|$, *i.e.,* the number of input variables involved in $S$. The order of interactions enables us to analyze the stability of interactions in a more precise way.

**Taking salient AND-OR interactions as primitive inference patterns encoded by a DNN.** Let us consider a DNN $v$ trained on a dataset $\mathcal{D}$. A well-trained DNN usually encodes complex interactions between input variables, including both AND interactions and OR interactions. Thus, Li & Zhang (2023a) proposed a method to simultaneously extract AND interactions $I_{\text{and}}(S|x)$ and OR interactions $I_{\text{or}}(S|x)$ from the network output $v(x) \in \mathbb{R}$.[4] Given any masked sample $x_S$ *w.r.t.* $S \subseteq N$, Li & Zhang (2023a) have proposed to learn a decomposition of network

---

[3]Each variable $x_i$ $(i \in N \setminus T)$ is masked by replacing its value to the baseline value $b_i$, *i.e.*, setting $(x_T)_i = b_i$. In this study, we adopt the setting of baseline values in (Ren et al., 2023c).

[4]For example, in the classification task, we set $v(x) = \log \frac{p_y(x)}{1-p_y(x)} \in \mathbb{R}$, where $p_y(x)$ denotes the predicted probability of the input $x$ belonging to the ground-truth category (*i.e.,* the $y$-th category).

outputs $v(x_S) = v_{\text{and}}(x_S) + v_{\text{or}}(x_S)$, towards the sparsest interactions. The $v_{\text{and}}(x_S)$ term is explained by AND interactions, and the $v_{\text{or}}(x_S)$ term is explained by OR interactions, subject to $I_{\text{and}}(\emptyset|x) = v_{\text{and}}(x_\emptyset) = v(\emptyset), I_{\text{or}}(\emptyset|x) = v_{\text{or}}(x_\emptyset) = 0$.

$$v(x_S) = v_{\text{and}}(x_S) + v_{\text{or}}(x_S), \quad v_{\text{and}}(x_S) = \sum\nolimits_{T \subseteq S} I_{\text{and}}(T|x), \quad v_{\text{or}}(x_S) = \sum\nolimits_{T \cap S \neq \emptyset} I_{\text{or}}(T|x) \tag{3}$$

*Although there is no theoretical analysis about whether such interactions fully align with human cognition, **a set of mathematical properties have been proven, which enable us to consider these interactions as primitive inference patterns encoded by the DNN.***

*(1) Universal matching.* Eq. (3) shows the universal matching property of AND-OR interactions. Given an input sample $x$, we can randomly mask $x$ and generate a total of $2^n$ different masked samples $\{x_S\}$ *w.r.t.* all $2^n$ subsets $S \subseteq N$. Then, given any one of the $2^n$ masked samples, the output of a DNN could always be well mimicked by the sum of both AND interactions and OR interactions.

*(2) Sparsity.* According to the definition of interactions, a model may encode up to $2^n$ AND interactions and $2^n$ OR interactions *w.r.t.* all subsets $S \subseteq N$. However, Ren et al. (2023d) have proven that in a well-trained DNN satisfying certain conditions,[1] only a small number of subsets $S$ have considerable AND interaction effects $I_{\text{and}}(S)$. **The AND interaction effects of most subsets $S$ are almost zero, *i.e.*, $I_{\text{and}}(S|x) \approx 0$.** Considering that the OR interactions can also be considered as AND interactions from another perspective (please see Appendix A for the proof), we can roughly consider that the OR interactions in a well-trained DNN also tend to be sparse.

*(3) Generalization power.* Li & Zhang (2023b) have discovered that in the classification task, the AND interactions have a good generalization power. Given DNNs trained for the same task, people can usually extract similar interactions on the same sample from different DNNs. Besides, the salient interactions are found discriminative for classification.

## 3.2 ILLUSTRATING THE INSTABILITY PROBLEM WITH INTERACTIONS

In this study, we discover some mathematical flaws in the above definition of AND-OR interactions. Although the AND-OR interaction satisfies the aforementioned properties of the *universal matching*, *sparsity*, and *generaization power*, it is still hard to consider these interactions as faithful primitive inference patterns encoded by the DNN.

**Non-uniqueness of interactions.** Given a DNN and an input sample, we find that we may extract different sets of AND-OR interactions from the same DNN, if we decompose the network output $v(x_S)$ in different ways. For example, let us consider a toy function $f(x) = 3(x_1 \wedge x_2) + (x_1 \vee x_2)$ with the input $x = [x_1, x_2]^T = [1, 1]^T$, $x_i \in \{0, 1\}$. $\wedge$ and $\vee$ denote the logical AND and OR operations, respectively. If we decompose $f(x_S) = 3((x_S)_1 \wedge (x_S)_2) + ((x_S)_1 \vee (x_S)_2) = v_{\text{and}}(x_S) + v_{\text{or}}(x_S)$ as $v_{\text{and}}(x_S) = 3((x_S)_1 \wedge (x_S)_2)$ and $v_{\text{or}}(x_S) = ((x_S)_1 \vee (x_S)_2)$, then the function output is explained as an AND interaction effect $I_{\text{and}}(\{x_1, x_2\}|x) = 3$ and an OR interaction effect $I_{\text{or}}(\{x_1, x_2\}|x) = 1$. Alternatively, we can also rewrite the same function as $f(x) = 2(x_1 \wedge x_2) + x_1 + x_2$. Then, $f(x_S)$ can be decomposed into $v_{\text{and}}(x_S) = 2((x_S)_1 \wedge (x_S)_2) + (x_S)_1 + (x_S)_2$ and $v_{\text{or}}(x_S) = 0$. In this case, we obtain three AND interaction effects (*i.e.*, $I_{\text{and}}(\{x_1, x_2\}|x) = 2, I_{\text{and}}(\{x_1\}|x) = 1$, and $I_{\text{and}}(\{x_2\}|x) = 1$), without any OR interaction effects. Thus, the extraction of AND-OR interactions highly depends on the decomposition of the output score.

**Sparsity does not mean stability.** Li & Zhang (2023a) propose to extract the sparsest interactions from each input sample, to tackle the non-uniqueness problem. Specifically, they decompose the network output $v(x_S)$ into $v_{\text{and}}(x_S) = 0.5 \cdot v(x_S) + \gamma_S$ and $v_{\text{and}}(x_S) = 0.5 \cdot v(x_S) - \gamma_S$, where $\{\gamma_S\}_{S \subseteq N}$ are a set of learnable parameters that determine the decomposition of output scores for AND and OR interactions. Then, they learn the parameters $\{\gamma_S\}$ by minimizing the following LASSO-like loss to obtain sparse interactions.

$$\min_{\{\gamma_S\}} \sum\nolimits_{S \subseteq N} [|I_{\text{and}}(S|x)| + |I_{\text{or}}(S|x)|] \tag{4}$$

However, the sparsest interactions are usually quite sensitive to small noises in the input. In fact, faithful interactions are expected to be stably extracted from different but similar samples. For example, given two similar images, similar interactions are supposed to be extracted from similar image patches in two images. Otherwise, if the extracted interactions are sensitive to small perturbations on image pixels for vision models or on word embeddings for language models, then these interactions may be overfitted to OOD features encoded by the DNN.

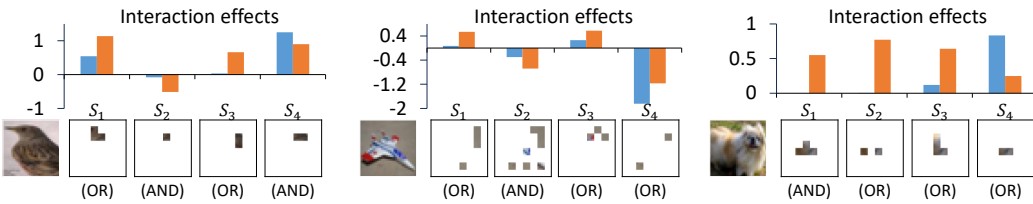

Figure 2: Comparison of numerical effects between interactions extracted from the original input sample and interactions extracted from the perturbed inputs.

Table 1: The ratio of OOD interactions in salient interactions extracted from the perturbed inputs.

| Dataset | census | | TV news | | CoLA | SST-2 | CIFAR-10 | |
| Model | MLP-5 | MLP-8 | MLP-5 | MLP-8 | LSTM | LSTM | ResNet-20 | LLaMA |
|---|---|---|---|---|---|---|---|---|
| (Li & Zhang, 2023a) | 24.48% | 27.88% | 22.37% | 26.59% | 6.87% | 9.37% | 17.82% | 16.39% |
| On-manifold interaction | **8.40%** | **11.55%** | **7.34%** | **14.03%** | **2.66%** | **2.57%** | **7.91%** | **8.46%** |

Besides, the difficulty of optimizing the above loss function is another reason for the instability of interactions. When we use different initializations of $\gamma_S$, we will obtain different sets of interactions with similar loss values. Such instability makes it hard to consider interactions as primitives for the network inference.

**Experimental verification of the instability of interactions.** To this end, we conducted an experiment to verify the instability of interactions. Using a ResNet-20 (He et al., 2016) trained on the CIFAR-10 dataset (Krizhevsky, 2012), we added small Gaussian noises ($\delta \sim \mathcal{N}(0, 0.1^2 I)$) on each input image $x$. We used Eq. (4) to compute the interactions in the original image and the interactions in the perturbed image, respectively. Figure 2 shows that the small perturbations on the image pixels caused us to extract different interactions. For some sets of input variables $S \subseteq N$, the interaction $I_{\text{and}}(S|x)$ and $I_{\text{or}}(S|x)$ extracted from the original input image based on Eq. (4) was fully different from that extracted from the perturbed input image ($I_{\text{and}}(S|x+\delta)$ or $I_{\text{or}}(S|x+\delta)$).

In addition, we also conducted another experiment to count the number of OOD interactions extracted from the perturbed inputs. Given the original input sample $x$, we perturbed $x$ by $K = 3$ Gaussian noises $\{\delta^{(k)}\}_{k=1,2,...,K}$ subject to $\delta^{(k)} \sim \mathcal{N}(0, 0.05^2 I)$, and obtained $K$ perturbed inputs $x^{(k)} = x + \delta^{(k)}$. Then, we computed the interaction effects ($I_{\text{and}}(S|x)$ and $I_{\text{or}}(S|x)$) in the original input $x$ and interaction effects ($I_{\text{and}}(S|x^{(k)})$ and $I_{\text{or}}(S|x^{(k)})$) in the perturbed inputs by optimizing Eq. (4), respectively. Let $\mathcal{S}_{\text{and}}^{(\text{ori})}(x) = \{S \subseteq N | |I_{\text{and}}(S|x)| \geq \tau_{\text{salient}}\}$ and $\mathcal{S}_{\text{or}}^{(\text{ori})}(x) = \{S \subseteq N | |I_{\text{or}}(S|x)| \geq \tau_{\text{salient}}\}$ denote salient AND interactions and OR interactions extracted from the original input sample, respectively. $\mathcal{S}_{\text{and}}^{(k)}(x) = \{S \subseteq N | |I_{\text{and}}(S|x^{(k)})| \geq \tau_{\text{salient}}\}$ and $\mathcal{S}_{\text{or}}^{(k)}(x) = \{S \subseteq N | |I_{\text{or}}(S|x^{(k)})| \geq \tau_{\text{salient}}\}$ denote salient AND-OR interactions extracted from the $k$-th perturbed input. We set $\tau_{\text{salient}} = 0.2 \cdot I_{\max}(x)$ for tabular datasets and set $\tau_{\text{salient}} = 0.05 \cdot I_{\max}(x)$ for language and image datasets, where $I_{\max}(x) = \max(\max_S(|I_{\text{and}}(S|x)|), \max_S(|I_{\text{or}}(S|x)|))$ represents the maximum absolute interaction effect over all AND-OR interactions extracted from $x$. The newly emerging salient interactions in $\mathcal{S}_{\text{and}}^{(k)} \setminus \mathcal{S}_{\text{and}}^{(\text{ori})}$ and $\mathcal{S}_{\text{or}}^{(k)} \setminus \mathcal{S}_{\text{or}}^{(\text{ori})}$ were considered OOD AND-OR interactions. The first row in Table 1 reports the average ratio of OOD interactions in salient interactions extracted from the perturbed inputs, *i.e.*, $\mathbb{E}_x \mathbb{E}_k \frac{|\mathcal{S}_{\text{and}}^{(k)}(x) \setminus \mathcal{S}_{\text{and}}^{(\text{ori})}(x)| + |\mathcal{S}_{\text{or}}^{(k)}(x) \setminus \mathcal{S}_{\text{or}}^{(\text{ori})}(x)|}{|\mathcal{S}_{\text{and}}^{(k)}(x)| + |\mathcal{S}_{\text{or}}^{(k)}(x)|}$. Under small input perturbations, the loss function in Eq. (4) usually generated many OOD interactions.

### 3.3 EXTRACTING ROBUST ON-MANIFOLD INTERACTIONS

Considering the discovered instability problem, we further propose to reformulate the objective function of extracting interactions, which aims to simultaneously constrain both the sparsity and stability (robustness) of interactions. We are inspired by the common hypothesis of the feature manifold in the high-dimensional feature space. Thus, we also hope to extract AND-OR interactions in a specific manifold, instead of obtaining noisy or unstable interactions, when we add small noises to the input.

We can understand on-manifold interactions as follows. Given the original input sample $x \in \mathbb{R}^n$ and $K$ Gaussian noises $\{\delta^{(k)}\}_{k=1,2,...,K}$ subject to $\delta^{(k)} \sim \mathcal{N}(0, \sigma^2 I) \in \mathbb{R}^n$, we obtain $K$ perturbed inputs $x^{(k)} = x + \delta^{(k)}$. Then, we extract $K$ sets of AND-OR interactions from the $K$ perturbed

inputs, respectively. If there is a large overlap between each pair of interaction sets, then we consider such interactions shared by different perturbed inputs to be on-manifold.

*In particular, on-manifold interactions mean that different perturbed inputs have similar sets of interactions, considering it has been proven that each input only has a few salient interactions. Noises on the input may mainly change the effect values of the shared interactions, rather than generate a fully new set of interactions.* We can consider the set/space of interactions as the manifold. In comparison, off-manifold interactions mean that we can obtain different sets of salient interactions under different noises.

In order to extract on-manifold interactions, we decompose the network output on each perturbed input as $v(x_S^{(k)}) = v_{\text{and}}(x_S^{(k)}) + v_{\text{or}}(x_S^{(k)})$, just like in (Li & Zhang, 2023a), where $v(x_S^{(k)})$ denotes the network output when we mask input variables in $N \setminus S$ in the $k$-th perturbed input $x^{(k)}$. $v_{\text{and}}(x_S^{(k)}) = 0.5 \cdot v(x_S^{(k)}) + \gamma_S^{(k)}$ and $v_{\text{or}}(x_S^{(k)}) = 0.5 \cdot v(x_S^{(k)}) - \gamma_S^{(k)}$. $\{\gamma_S^{(k)}\}_{S \subseteq N}$ denotes the set of learnable parameters for the $k$-th perturbed input, $1 \leq k \leq K$. In this way, the most straightforward way of extracting sparse interactions shared by different perturbed inputs while penalizing OOD interactions can be formulated as follows.

$$\min_{\{\gamma_S^{(k)}\}} \sum_{S \subseteq N} \left[ \|\mathcal{I}_{\text{and},S}\|_\infty + \|\mathcal{I}_{\text{or},S}\|_\infty \right] \tag{5}$$

where $\mathcal{I}_{\text{and},S} = [I_{\text{and}}(S|x^{(1)}), I_{\text{and}}(S|x^{(2)}), \ldots, I_{\text{and}}(S|x^{(K)})]^T \in \mathbb{R}^K$ denotes the vector of all AND interaction effects of the subset $S$ extracted from $K$ perturbed inputs, and $\mathcal{I}_{\text{or},S} = [I_{\text{or}}(S|x^{(1)}), I_{\text{or}}(S|x^{(2)}), \ldots, I_{\text{or}}(S|x^{(K)})]^T \in \mathbb{R}^K$ denotes the vector of all OR interaction effects. $\|\cdot\|_\infty$ represents the $\ell_\infty$-norm of the vector. This objective extends the LASSO loss by applying the $\ell_\infty$-norm to penalize the interaction with the most significant numerical effect, which also penalizes the potential OOD interactions. In other words, if the subset $S$ has a significant interaction effect in a perturbed input, then the same set is also expected to have considerable interaction effects in other perturbed inputs. On the other hand, non-salient interactions are supposed to also keep silent on all $K$ perturbed inputs. In this way, interactions extracted from $K$ perturbed inputs are pushed to be similar and are considered on-manifold interactions.

**Sharing decomposition.** Directly optimizing Eq. (5) is difficult, so we also apply a trick to boost the computational efficiency. The $K$ different perturbed inputs are supposed to have similar decompositions of output scores. Thus, we slightly revise the setting to $v_{\text{and}}(x_S^{(k)}) = 0.5 \cdot v(x_S^{(k)}) + \bar{\gamma}_S + \gamma_S^{(k)}$ and $v_{\text{or}}(x_S^{(k)}) = 0.5 \cdot v(x_S^{(k)}) - \bar{\gamma}_S - \gamma_S^{(k)}$, where $\bar{\gamma}_S$ represents the common decomposition of the output score shared by $K$ perturbed inputs, and $\gamma_S^{(k)}$ represents the distinct term for each perturbed input. In particular, we limit the distinct term $\gamma_S^{(k)}$ to a small range, *i.e.*, $|\gamma_S^{(k)}| \leq \tau$, subject to $\tau = 0.02 \cdot \mathbb{E}_k[|v(x^{(k)}) - v(x_\emptyset^{(k)})|]$. During the optimization, we clamp the value of $\gamma_S^{(k)}$ to be in the range $[-\tau, \tau]$ after the update of $\gamma_S^{(k)}$ in each iteration.

**Towards the redundancy of interactions.** The aforementioned non-uniqueness problem may generate redundant interactions during the loss minimization in Eq.( 5). To be precise, for each subset $S \subseteq N$, the loss exclusively penalizes the most salient interaction effect $\max_k |I_{\text{and}}(S|x^{(k)})|$ and $\max_k |I_{\text{or}}(S|x^{(k)})|$ over $K$ perturbed inputs, but it does not constrain interaction effects extracted from the other $K - 1$ perturbed inputs. Thus, the algorithm may find a short-cut solution that uses unconstrained interactions to represent the effect of the salient interaction, which actually transfers the penalty to unconstrained interactions. This will generate redundant and non-transferable interactions. Please see Appendix C.1 for details. Therefore, we further revise the above loss function to incorporate some penalties of non-salient interaction effects.

$$\min_{\{\bar{\gamma}_S, \gamma_S^{(k)}\}} \mathcal{L} = \mathcal{L}_{\text{salient}} + \lambda \mathcal{L}_{\text{non-salient}},$$
$$\mathcal{L}_{\text{salient}} = \sum_{S \subseteq N} \left[ \sum_{k \in \Omega_{\text{and}}(S)} |I_{\text{and}}(S|x^{(k)})| + \sum_{k \in \Omega_{\text{or}}(S)} |I_{\text{or}}(S|x^{(k)})| \right], \tag{6}$$
$$\mathcal{L}_{\text{non-salient}} = \sum_{S \subseteq N} \left[ \sum_{k \in \{1,2,\ldots,K\} \setminus \Omega_{\text{and}}(S)} |I_{\text{and}}(S|x^{(k)})| + \sum_{k \in \{1,2,\ldots,K\} \setminus \Omega_{\text{or}}(S)} |I_{\text{or}}(S|x^{(k)})| \right]$$

In the above loss, the $L_{\text{salient}}$ term extends the penalty for the maximum interaction in Eq. (5) to the set of top-$r$ most salient AND-OR effects among all $K$ interaction effects for each subset $S \subseteq N$. $\Omega_{\text{and}}(S)$ and $\Omega_{\text{or}}(S)$ denote the set of top-$r$ salient AND interactions and OR interactions of $S$, respectively. In addition, we also add small penalties $L_{\text{non-salient}}$ to other $K - r$ non-salient interaction effects, which are weighted by $\lambda$. In experiments, we set the number of perturbed inputs $K = 3$ and the number of salient interactions $r = 1$. We set the weight $\lambda = 0.1$.

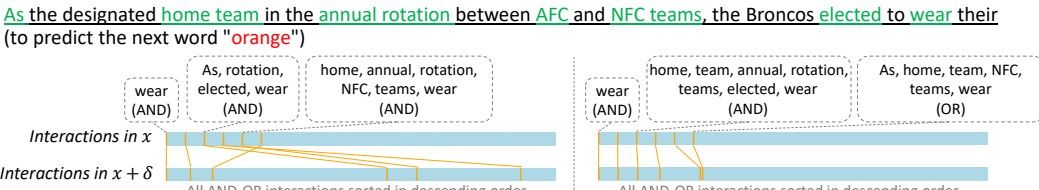

Figure 3: Illustrating the stability of interactions with the 1st, 100th, 200th, 300th 400th, and 500th largest effects, which were extracted from LLaMA. To reduce computational cost, we selected green words as input variables and took other black words as the constant background. The blue axes represent all AND-OR interactions sorted in the descending order. Orange lines show how the 1st, 100th, 200th, 300th 400th, and 500th-ranked interactions were ranked in the perturbed input $(x + \delta)$.

**Modeling noises.** There is a typical problem with interactions, *i.e.,* small noises in the network output may significantly affect the extraction of interactions, especially for complex interactions between lots of variables. Specifically, let us suppose the network output $v(x_S^{(k)})$ contains a tiny Gaussian noise $\epsilon_S^{(k)} \sim N(0, \sigma_\epsilon^2)$, *i.e.,* $v(x_S^{(k)}) \leftarrow v(x_S^{(k)}) + \epsilon_S^{(k)}$. Then, we can prove that $I_{\text{and}}(S|x^{(k)}, v)$ follows a Gaussian distribution with the variance $\sigma_I^2 = 2^{|S|} \cdot \sigma_\epsilon^2$. Please see Appendix B for the proof. The large variance $\sigma_I^2$ indicates that small noises on network outputs may significantly change the extracted interaction effects, especially for high-order interactions.

Thus, we propose to also learn to remove the noise term $\epsilon_S^{(k)}$ from the computation of AND-OR interactions. We rewrite the decomposition as $v_{\text{and}}(x_S^{(k)}) = 0.5 \cdot (v(x_S^{(k)}) - \epsilon_S^{(k)}) + \bar{\gamma}_S + \gamma_S^{(k)}$ and $v_{\text{or}}(x_S^{(k)}) = 0.5 \cdot (v(x_S^{(k)}) - \epsilon_S^{(k)}) + \bar{\gamma}_S + \gamma_S^{(k)}$. Thus, we simultaneously learn the parameters $\{\epsilon_S^{(k)}\}$, $\{\bar{\gamma}_S\}$, and $\{\gamma_S^{(k)}\}$ by minimizing the loss function in Eq. (6). Similar to $\{\gamma_S^{(k)}\}$, the values of $\{\epsilon_S^{(k)}\}$ are also constrained in $[-\tau, \tau]$ where $\tau = 0.02 \cdot \mathbb{E}_k[|v(x^{(k)}) - v(x_\emptyset^{(k)})|]$.

## 4 EXPERIMENTS

**Datasets and models.** In this section, we used Eq. (6) to extract on-manifold interactions in tabular datasets, language datasets, and image datasets. We followed (Ren et al., 2023c) to train a 5-layer MLP (namely *MLP-5*) and an 8-layer MLP (namely *MLP-8*) on two tabular datasets, including the UCI census income dataset (Dua & Graff, 2017) (namely *census*) and the UCI TV News channel commercial detection dataset (Dua & Graff, 2017) (namely *TV news*). For language data, we followed the settings in (Ren et al., 2023c) to train two three-layer unidirectional LSTM (Hochreiter & Schmidhuber, 1997) models on the SST-2 dataset (Socher et al., 2013) and on the CoLA dataset (Warstadt et al., 2018), respectively. Besides, we also extracted interactions from the pre-trained 7B LLaMA model (Touvron et al., 2023) on the sentence completion task. For image data, we trained a ResNet-20 (He et al., 2016) on the CIFAR-10 (Krizhevsky, 2012) dataset.

For each input sample $x$ in either a tabular dataset or an image dataset, we randomly sampled $K = 3$ noise vectors $\{\delta^{(k)}\}$ from the Gaussian distribution $\mathcal{N}(0, \sigma^2 I)$ with $\sigma = 0.05$, and then added $\delta^{(k)}$ to the input $x$ to obtain the perturbed input $x^{(k)} = x + \delta^{(k)}$. Besides, we used image patches in each input image in the CIFAR-10 dataset, which were sampled by following (Ren et al., 2023c), as input variables to extract interactions. For each input sentence of LSTM models, we also sampled $K = 3$ noise vectors from $\mathcal{N}(0, \sigma^2 I)$ with $\sigma = 0.05$, and we added noises to embeddings of input tokens/words. For the prompt sentence fed into the large language model LLaMA, we sampled noises from the Gaussian distribution with $\sigma = 0.01$, and added them to embeddings of input tokens. If the length of the sentence is larger than 10, we randomly selected 10 tokens as input variables. Figure 3 shows an example of interactions extracted from the LLaMA model. On-manifold interactions extracted from the original input and those extracted from the perturbed input were similar, while the extracted traditional interactions (Li & Zhang, 2023a) were not similar.

### 4.1 TRADITIONAL INTERACTIONS V.S. ON-MANIFOLD INTERACTIONS

**Comparison of sparsity.** We conducted experiments to compare the sparsity of the on-manifold interactions extracted using the loss function in Eq. (6), with the sparsity of the traditional interac-

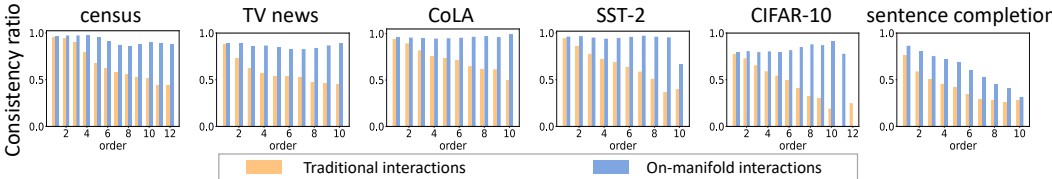

Figure 4: Similarity of interaction effects between interactions from different perturbed inputs

Figure 5: Ratio of consistent interactions. On-manifold interactions exhibited much higher consistency ratios than traditional interactions, especially for the comparison of high-order interactions.

tions extracted by (Li & Zhang, 2023a). To extract on-manifold interaction from the original input, we jointly extracted $K + 1$ sets of interactions from the original input and other $K$ perturbed inputs ($K = 3$), respectively. In this way, the on-manifold interactions extracted from the original input sample were compared to traditional interactions extracted from the same input sample. Figure 9 in Appendix C.2 shows that although the proposed on-manifold interaction brought in a new loss for the stability of interactions, it did not hurt the sparsity of interactions.

Furthermore, we proposed the following two metrics to quantify the stability of interaction effects extracted under different input perturbations.

**Metric 1: similarity between interaction effects under different input perturbations.** Given each input sample, let us extract $K$ sets of interactions of the $m$-th order when we added $K$ different perturbations to the input sample. We extended the Jaccard similarity to compute the similarity between $K$ sets of m-order interactions, $similarity_m = \left[ \frac{\sum_{|S|=m} \sum_{\text{op} \in \{\text{and}, \text{or}\}} \min_k(|I_{\text{op}}(S|x^{(k)})|)}{\sum_{|S|=m} \sum_{\text{op} \in \{\text{and}, \text{or}\}} \max_k(|I_{\text{op}}(S|x^{(k)})|)} \right]$. Figure 4 reports the average value over similarities computed on different input samples. To construct a competing baseline, we also extracted traditional interaction effects based on Eq. (4) on each perturbed input, so that we could also compute similarities from these interaction effects.

Figure 4 shows that under different input perturbations, traditional interactions extracted by (Li & Zhang, 2023a) usually presented much lower similarity than on-manifold interactions, especially for high-order (complex) interactions.

**Metric 2: consistency of interactions in the original input and interactions in perturbed inputs.** Given an input image $x$, we jointly extracted interaction effects in the original input and other $K$ perturbed inputs. Then, we selected the top 400 salient interactions with the largest values of $|I_{\text{and}}(S|x)|$ or $|I_{\text{or}}(S|x)|$ in the original input, denoted by $\mathcal{S}_{\text{and}}(x)$ and $\mathcal{S}_{\text{or}}(x)$, respectively. For each salient interaction, we examined whether its corresponding effects in perturbed inputs were consistent with its effect value $I_{\text{and}}(S|x)$ or $I_{\text{and}}(S|x)$ in the original input. For example, let us focus on a salient AND interaction $I_{\text{and}}(S|x)$. We considered the interaction effect $I_{\text{and}}(S|x^{(k)})$ to be consistent with $I_{\text{and}}(S|x)$ in the original input if and only if $I_{\text{and}}(S|x^{(k)})$ contained more than 85% signals in $I_{\text{and}}(S|x)$, *i.e.,* $I_{\text{and}}(S|x^{(k)})/I_{\text{and}}(S|x) \geq 0.85$. In this way, we computed the ratio of interaction effects $I_{\text{and}}(S|x^{(k)})$ in all $K$ perturbed inputs that were consistent with the salient interaction $I_{\text{and}}(S|x)$ in the original input, denoted by $\alpha_{\text{and}, S} = \frac{1}{K} \sum_k \mathbb{1}(I_{\text{and}}(S|x^{(k)})/I_{\text{and}}(S|x) \geq 0.85)$. Then, Figure 5 reports the average consistency ratio for all $m$-order interactions among the all top 400 salient interactions in $\mathcal{S}_{\text{and}}(x)$ and $\mathcal{S}_{\text{or}}(x)$, *i.e.,* $\mathbb{E}_{\text{op} \in \{\text{and}, \text{or}\}} \mathbb{E}_{|S|=m, S \in \mathcal{S}_{\text{op}}(x)}[\alpha_{\text{op}, S}]$. Just like the comparison of the similarity in Figure 4, we also used Eq. (4) to extract traditional interaction effects of the $K + 1$ inputs and computed their consistency ratio as a baseline. Figure 5 shows that on-manifold interaction effects exhibited much higher consistency ratios than traditional interaction effects.

### 4.2 EXPLAINING ADVERSARIAL VULNERABILITY

Faithfully extracting on-manifold interactions under different noises provides us with a new perspective on explaining the adversarial vulnerability of DNNs. Because the network output can be

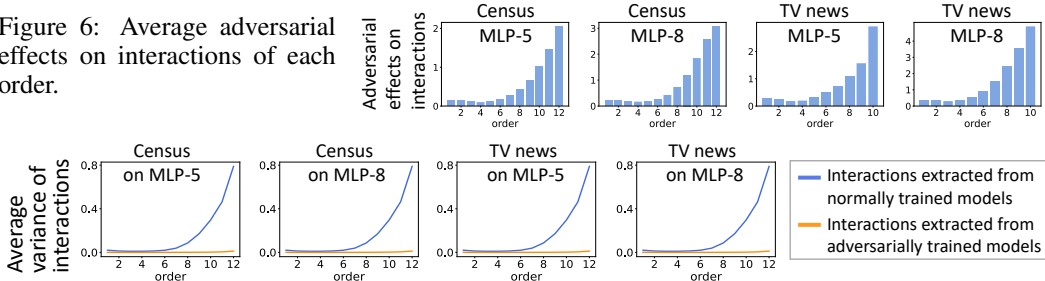

Figure 6: Average adversarial effects on interactions of each order.

Figure 7: Variance (sensitivity) of interaction effects caused by different input perturbations. We compared interaction effects extracted from normally-trained DNNs and adversarially-trained DNNs. Adversarial training significantly reduced the variance of interaction effects.

disentangled into effects of different interactions, *i.e.*, $v(x) = \sum_{S \subseteq N} I_{\mathrm{and}}(S|x) + \sum_{S \subseteq N} I_{\mathrm{or}}(S|x)$ according to Eq. (3), we can decompose the overall adversarial vulnerability $\Delta v(x) = v(x^{\mathrm{adv}}) - v(x)$ into sensitivities of different interactions. *I.e.*, $\Delta v(x) = \sum_{S \subseteq N} \Delta I_{\mathrm{and}}(S|x) + \sum_{S \subseteq N} \Delta I_{\mathrm{or}}(S|x)$, where $x^{\mathrm{adv}}$ denotes the adversarial example. $\Delta I_{\mathrm{and}}(S|x) = I_{\mathrm{and}}(S|x^{\mathrm{adv}}) - I_{\mathrm{and}}(S|x)$ represents the change of the numerical effect of the AND interaction in $S$ caused by the adversarial perturbation. To this end, previous studies have explored similar explanations for adversarial vulnerability. For example, Ren et al. (2021) have used the sensitivity of multi-order bivariate interactions (Zhang et al., 2021) to explain the adversarial robustness of DNNs.

Compared to explanations based on multi-order bivariate interaction, the AND-OR interactions in this paper have been proven to have the sparsity, universal matching property, and generalization power. Moreover, according to Figures 4 and 5, the on-manifold interactions proposed in this paper are more robust to small perturbations on the input. Thus, such on-manifold interactions are supposed to be more convincing metrics to explain primitive adversarial effects.

Thus, we conducted experiments to compute the influence of adversarial perturbations on the on-manifold interaction effects. Given each input sample $x$, we extracted on-manifold interaction effects from both the original sample $x$ and its adversarial example $x^{\mathrm{adv}}$. Let $\Delta I_{\mathrm{and}}(S|x) = I_{\mathrm{and}}(S|x^{\mathrm{adv}}) - I_{\mathrm{and}}(S|x)$ denote the adversarial effect on $I_{\mathrm{and}}(S|x)$, and let $\Delta I_{\mathrm{or}}(S|x) = I_{\mathrm{or}}(S|x^{\mathrm{adv}}) - I_{\mathrm{or}}(S|x)$ denote the adversarial effect on $I_{\mathrm{or}}(S|x)$. We further averaged adversarial effects of interactions of the same order, *i.e.*, $\mathbb{E}_{|S|=m}\mathbb{E}_{\mathrm{op} \in \{\mathrm{and,or}\}}|\Delta I_{\mathrm{op}}(S|x)|$. Figure 6 shows that high-order (complex) interactions are more vulnerable to adversarial perturbations than low-order (simple) interactions.

### 4.3 ADVERSARIAL TRAINING ENHANCES THE STABILITY OF INTERACTIONS

In this section, we found that adversarial training enhanced the stability of AND-OR interactions. We adversarially trained MLPs on the census and TV news datasets, respectively. The adversarial settings and accuracy of models are reported in Appendix C.3. Then, we extracted on-manifold interactions from the adversarially trained MLPs. Given $K$ sets of interaction effects extracted under $K$ perturbed inputs, we computed the variance of each interaction over $K$ perturbed versions, denoted by $\mathrm{var}_{\mathrm{and}}(S|x)$ and $\mathrm{var}_{\mathrm{or}}(S|x)$. Figure 7 reports the average variance of interactions of each order $m$, *i.e.*, $\mathbb{E}_{|S|=m}\mathbb{E}_{\mathrm{op} \in \{\mathrm{and,or}\}}[\mathrm{var}_{\mathrm{op}}(S|x)]$. High-order (complex) interactions were more sensitive to perturbations than low-order (simple) interactions. Moreover, the adversarial training significantly reduced the variance of interaction effects and enhanced the stability of interactions.

## 5 CONCLUSIONS

In this study, we have proposed a method to extract on-manifold interactions between input variables of DNNs. Compared to traditional interactions, the proposed on-manifold interactions are more stable (robust) to small perturbations on the input. Experimental results have demonstrated the stability of on-manifold interactions extracted from different perturbed inputs. Therefore, we consider the on-manifold interactions as a more faithful approach to understanding primitive inference patterns encoded by the DNN. Besides, the on-manifold interactions have also been used to explain the effects of adversarial perturbations on the network output and the effects of adversarial training.

ETHIC STATEMENT

This paper aims to extract stable and on-manifold interactions encoded by the DNN to better explain the DNN. We discover that although the AND-OR interactions have some good mathematical properties, the interactions extracted by the previous method are unstable to small noises on the input. Such instability makes people cannot rigorously take the extracted interactions as primitive inference patterns of the DNN. Thus, we propose a method to extract on-manifold interactions, which are relatively stable to input noises. Furthermore, the extracted on-manifold interactions can help us understand the effects of adversarial perturbations on the network output. There are no ethical issues with this paper.

REPRODUCIBILITY STATEMENT

We have provided proofs for the theoretical results of this study in Appendix A and Appendix B. We have also provided experimental details in Section 4 and Appendix C. Furthermore, we will release the code when the paper is accepted.

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

# A  RELATIONSHIP BETWEEN AND INTERACTIONS AND OR INTERACTIONS

In this section, we provide a further discussion on understanding the relationship between AND and OR interactions.

Given a DNN $v(\cdot)$ and an input sample $x \in \mathbb{R}^n$, let $b \in \mathbb{R}^n$ denote the baseline values input variables, which represent the masked states of variables in the input. Then, if the variables in $L \subseteq N$ are preserved and other variables are masked, the masked input $x_L$ is defined as follows.

$$(x_L)_i = \begin{cases} x_i, & i \in L \\ b_i, & i \notin L \end{cases} \tag{7}$$

Based on the above definition of masked inputs, the AND interaction can be computed as $I_{\text{and}}(S|x) = \sum_{L \subseteq S}(-1)^{|S|-|L|}v_{\text{and}}(x_L)$. The OR interaction between variables in $x$ is computed as $I_{\text{or}}(S|x) = -\sum_{L \subseteq S}(-1)^{|S|-|L|}v_{\text{or}}(x_{N \setminus L})$. To simplify the analysis, we assume $v_{\text{and}}(\cdot) = v_{\text{or}}(\cdot) = 0.5v(\cdot)$.

Conversely, if we consider $b$ as the input sample and take $x$ as baseline values of input variables in $b$, the masked input $b_L$ is defined as follows.

$$(b_L)_i = \begin{cases} b_i, & i \in L \\ x_i, & i \notin L \end{cases} \tag{8}$$

According to Eq. (7) and Eq. (8), we have $x_{N \setminus L} = b_L$. Therefore, we can rewrite the OR interaction as follows.

$$\begin{aligned} I_{\text{or}}(S|x) &= -\sum_{L \subseteq S}(-1)^{|S|-|L|}v_{\text{or}}(x_{N \setminus L}) \\ &= -\sum_{L \subseteq S}(-1)^{|S|-|L|}v_{\text{or}}(b_L) \\ &= -\sum_{L \subseteq S}(-1)^{|S|-|L|}v_{\text{and}}(b_L) \\ &= -I_{\text{and}}(S|b) \end{aligned} \tag{9}$$

Therefore, the OR interaction can be viewed as a special AND interaction by considering original variable values $x$ as masked states and taking the masked states $b$ as normal values of the variables.

# B  NOISES IN NETWORK OUTPUTS SIGNIFICANTLY AFFECT THE INTERACTION EFFECTS

Let us assume that the network output $v(x_S)$ on the masked input $x_S$ has a small noise $\epsilon_S$, *i.e.,* $\forall S \subseteq N, v(x_S) = u(x_S) + \epsilon_S$ with $\epsilon_S \sim \mathcal{N}(0, \sigma_\epsilon^2)$, where $u(x_S)$ denotes the precise network output without any noises. Then, we prove that small noises in the network output can significantly change the interaction effect.

**Theorem 1.** *Suppose the network outputs contain small noises,* i.e., $\forall S \subseteq N, v(x_S) = u(x_S) + \epsilon_S$ *with* $\epsilon_S \sim \mathcal{N}(0, \sigma_\epsilon^2)$. *Then, the AND interaction* $I_{and}(S|x, v) = I_{and}(S|x, u) + \sum_{T \subseteq S}(-1)^{|S|-|T|}\epsilon_T$ *is proven to follow a Gaussian distribution,* $I_{and}(S|x, v) \sim \mathcal{N}(\mu_{I_{and}} = I_{and}(S|x, u), \sigma_{I_{and}}^2 = 2^{|S|}\sigma_\epsilon^2)$. *Similarly, the OR interaction* $I_{or}(S|x, v) = I_{or}(S|x, u) + \sum_{T \subseteq S}(-1)^{|S|-|T|}\epsilon_T$ *is also proven to follow a Gaussian distribution,* $I_{or}(S|x, v) \sim \mathcal{N}(\mu_{I_{or}} = I_{or}(S|x, u), \sigma_{I_{or}}^2 = 2^{|S|}\sigma_\epsilon^2)$.

*Proof.* The proof for OR interactions is similar to the proof for AND interactions. Let us mainly focus on the AND interaction $I_{\text{and}}(S|x, v)$, whose mean value can be written as

$$\begin{aligned} \mu(I_{\text{and}}(S|x, v)) &= \mu\left(I_{\text{and}}(S|x, u) + \sum_{T \subseteq S}(-1)^{|S|-|T|}\epsilon_T\right) \\ &= \mu\left(I_{\text{and}}(S|x, u)\right) + \mu\left(\sum_{T \subseteq S}(-1)^{|S|-|T|}\epsilon_T\right) \\ &= I_{\text{and}}(S|x, u) + \sum_{T \subseteq S}(-1)^{|S|-|T|}\mu(\epsilon_T) \quad // \quad \text{noises on different terms are independent.} \\ &= I_{\text{and}}(S|x, u) \quad // \quad \forall T \subseteq S, \mu(\epsilon_T) = 0. \end{aligned} \tag{10}$$

Figure 8: Comparison of AND-OR interactions extracted by using Eq. (5) (w/o optimizing $Loss_{\text{non-salient}}$) and AND-OR interaction extracted by using Eq. (6) (with optimizing $Loss_{\text{non-salient}}$). Eq. (5) extracted many redundant OR interactions.

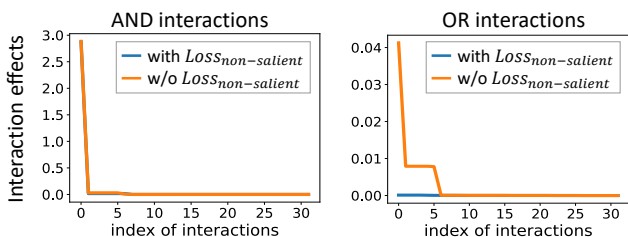

The variance of the AND interaction $I_{\text{and}}(S|x, v)$ can be written as follows.

$$Var(I_{\text{and}}(S|x, v)) = Var\left(I_{\text{and}}(S|x, u) + \sum_{T \subseteq S}(-1)^{|S|-|T|}\epsilon_T\right)$$
$$= Var\left(I_{\text{and}}(S|x, u)\right) + Var\left(\sum_{T \subseteq S}(-1)^{|S|-|T|}\epsilon_T\right). \tag{11}$$

This is because the AND interaction $I_{\text{and}}(S|x, u)$ and the Gaussian noise $\epsilon_T$ are independent. In the above equation, the first term is a constant with a variance of zero. For the second term, because the Gaussian noises $\forall S \subseteq N, \epsilon_S \sim \mathcal{N}(0, \sigma_\epsilon^2)$ are *i.i.d.*, then the variance $Var\left(\sum_{T \subseteq S}(-1)^{|S|-|T|}\epsilon_T\right)$ can be written as follows.

$$Var\left(\sum_{T \subseteq S}(-1)^{|S|-|T|}\epsilon_T\right) = Var(\epsilon_{T_1}) + Var(\epsilon_{T_2}) + \cdots + Var(\epsilon_{T_{2|S|}})$$
$$= 2^{|S|} \cdot \sigma_\epsilon^2. \quad // \quad \text{there are } 2^{|T|} \text{ subsets } T \subseteq S \text{ in total.} \tag{12}$$

Thus, we have $I_{\text{and}}(S|x, v) \sim \mathcal{N}(\mu_{I_{\text{and}}} = I_{\text{and}}(S|x, u), \sigma_{I_{\text{and}}}^2 = 2^{|S|} \cdot \sigma_\epsilon^2)$. $\qquad \square$

## C  IMPLEMENTATION DETAILS IN EXPERIMENTS

### C.1  ABOUT THE REDUNDANCY OF INTERACTIONS IN EQ. (5)

In this section, we conducted an experiment to show that the loss function in Eq. (5) generated some redundant interactions. In Eq.( 5), for each subset $S \subseteq N$, the loss exclusively penalizes the most salient interaction effect $\max_k |I_{\text{and}}(S|x^{(k)})|$ and $\max_k |I_{\text{or}}(S|x^{(k)})|$ over $K$ perturbed inputs without constraining interaction effects extracted from the other $K-1$ perturbed inputs. Therefore, the unconstrained interactions may take on some penalized effects in the most salient interaction. This will generate redundant and non-transferable interactions.

Let us consider a toy example. Given the function $f(x) = 3x_1 x_2 x_3 x_4 x_5$ with the input $x = [1, 1, 1, 1, 1]^T$, it is supposed to contain only one AND interaction $I_{\text{and}}(\{x_1, x_2, x_3, x_4, x_5\}|x) = 3$. We perturbed the input $x$ with $K = 3$ noises sampled from the Gaussian distribution $\mathcal{N}(0, \sigma^2 I)$ with $\sigma = 0.05$. Then, we jointly extracted $K+1$ sets of interactions from the original input and the perturbed inputs using Eq. (5) (which is the same as only optimizing $Loss_{\text{salient}}$ without optimizing $Loss_{\text{non-salient}}$ in Eq. (6)). For comparison, we also extracted interactions using Eq. (6) with $\lambda = 0.5$. Figure 8 shows the distribution of absolute values of the AND-OR interaction effects extracted from the original input, which were sorted in a descending order. While two sets of AND interaction effects were similar, Eq. (5) without $Loss_{\text{non-salient}}$ extracted many non-zero OR interactions, which were noisy and redundant.

### C.2  SPARSITY OF ON-MANIFOLD INTERACTIONS

We compared the sparsity of the on-manifold interactions extracted using the loss function in Eq. (6), with the sparsity of the traditional interactions extracted by (Li & Zhang, 2023a) in Figure 9. To extract on-manifold interaction from the original input, we jointly extracted $K+1$ sets of interactions from the original input sample and other $K$ perturbed inputs, respectively ($K = 3$). In this way, the on-manifold interactions extracted from the original input sample were compared to traditional interactions extracted from the same input sample. Figure 9 shows strength ($|I_{\text{and}}(S|x)|$ and $|I_{\text{or}}(S|x)|$) of all AND-OR interactions extracted from 50 input samples, which were sorted in a descending order.

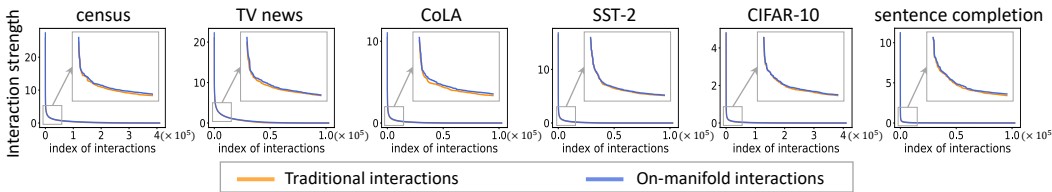

Figure 9: Visualizing strength of AND-OR interactions ($|I_{and}(S|x)|$ and $|I_{or}(S|x)|$) extracted from 50 input samples in a descending order. The proposed on-manifold interactions (blue) have similar sparsity as traditional interactions extracted by (Li & Zhang, 2023a) (orange).

### C.3  ADVERSARIAL TRAINING

In Section 4.3, we adversarially trained two MLP-5s and MLP-8s on the census and TV news datasets, respectively. We used $\ell_\infty$ PGD attack to generate adversarial examples for training. The attacking budget was set as $\epsilon = 0.1$. For the census dataset, we attacked each input for 20 steps with the step size of $0.01$. For the TV news dataset, we attacked each input for 10 steps with the step size of $0.02$. Table 2 reports the accuracy of normally trained models and adversarially trained models on clean samples and adversarial examples, respectively.

Table 2: The accuracy of normally trained models and adversarially trained models.

|  |  | Clean sample | | Adversarial example | |
|---|---|---|---|---|---|
|  |  | train set | test set | train set | test set |
| the census dataset | Normally trained MLP-5 | 89.93% | 84.12% | 68.07% | 63.96% |
|  | Adversarially trained MLP-5 | 85.49% | 84.87% | 84.06% | 83.18% |
|  | Normally trained MLP-8 | 93.01% | 83.16% | 62.21% | 57.87% |
|  | Adversarially trained MLP-8 | 85.70% | 84.73% | 84.24% | 82.73% |
| the TV news dataset | Normally trained MLP-5 | 91.19% | 81.64% | 33.47% | 31.12% |
|  | Adversarially trained MLP-5 | 81.18% | 80.16% | 70.30% | 68.38% |
|  | Normally trained MLP-8 | 94.62% | 79.70% | 31.89% | 29.60% |
|  | Adversarially trained MLP-8 | 81.04% | 79.80% | 70.39% | 68.52% |

