# OpenReview forum: "Extracting Robust On-Manifold Interactions Encoded by Neural Networks"
_ICLR.cc/2024/Conference — ICLR 2024 Conference Withdrawn Submission_

### Official Review · Reviewer_DNgx · 2023-11-01

**Soundness:** 3 good
**Presentation:** 3 good
**Contribution:** 2 fair
**Rating:** 6
**Confidence:** 2

**Summary:**

This paper targets the problem of finding the relationship between individual features in each input data example according to their response on trained deep neural networks. This relationship is defined as interactions between input variables indicating whether a subset of variables only collectively activate a concept (AND relationship) or individually activate a concept (OR relationship). The proposed method is built upon the objective function of an existing work, which tackles this problem by formulating a decomposed network output with learnable parameters indicating possible AND OR interactions of subsets. Specifically, this optimization objective is further improved in this work with two new modifications that take into consideration the consistency of the learned interactions from multiple noised inputs.

**Strengths:**

The problem of extracting explainable interactions in input features is interesting.

The proposed method is well motivated and reasonable.

Experiments on multiple different types of datasets demonstrate the effectiveness of the proposed method in increasing the robustness of discovered interactions.

**Weaknesses:**

Although the problem is interesting, the proposed method is built upon the idea of consistency outputs for different noised versions of input data. This technique has been widely used in deep learning research, for example for images, it has been shown that training with augmented inputs under loss functions tailored to consistency can greatly improve robustness. Many other augmentation techniques, including mixing examples besides adding noise, have also been investigated in this area. Therefore, it is not technically novel or exciting when applying this technique to other optimization objectives for deep neural network including explainable learning.

Other questions:

In Eq. (4), the variable in optimization is $\gamma$ but they are not shown in the equation on the right-hand side of $\min_{\gamma}$. I understand that $\gamma$ determines the values of some variables in the equation, it would be better to indicate how the formula is calculated from $\gamma$ to improve self-consistency.

**Questions:**

See weakness part.

---

### Official Review · Reviewer_5igS · 2023-11-02

**Soundness:** 2 fair
**Presentation:** 3 good
**Contribution:** 2 fair
**Rating:** 3
**Confidence:** 3

**Summary:**

The paper shows that interactions can be roughly considered primitive inference patterns encoded by a DNN, given that a small number of interactions can accurately explain the network outputs on any randomly masked samples. The discussion on the stability of interactions motivates the authors to utliize the on-manifold interactions.

**Strengths:**

The on-manifold interactions have meaningful importance.

The paper is completed.

The paper writing has no grammar mistakes.

**Weaknesses:**

The motivation is not clearly illustrated.

The prominent contribution is not well-presented. No core contribution is shown in the method.

The utilization of AND and OR interactions is not well constructed.

**Questions:**

1. Since the paper gives an example of the image and interactions from image patches, how did the paper construct the interactions on image patches? All interactions shown in the paper are constructed on sentences.

2. Is the instability problem with interactions treated as the main contribution? However, I see little theoretical content but more experimental evaluation of this instability in the method part.

3. The extraction of robust on-manifold interactions is conducted under K sets, and shows little novel analysis and discussion.

Actually, I am not familiar with this task. But I think the authors would better clarify the definition and advantages of on-manifold interactions, compared to other high-level interaction works. Besides, the robustness of on-manifold interactions should be clearly discussed in theory and experiments.

---

### Official Review · Reviewer_yWZn · 2023-11-08

**Soundness:** 2 fair
**Presentation:** 1 poor
**Contribution:** 1 poor
**Rating:** 3
**Confidence:** 2

**Summary:**

The paper builds upon previous works, namely:

[1] Ren, J., Li, M., Chen, Q., Deng, H. and Zhang, Q., 2023. Defining and quantifying the emergence of sparse concepts in dnns. In Proceedings of the IEEE/CVF Conference on Computer Vision and Pattern Recognition (pp. 20280-20289).

[2] Ren, Q., Gao, J., Shen, W. and Zhang, Q., 2023. Where We Have Arrived in Proving the Emergence of Sparse Symbolic Concepts in AI Models. arXiv preprint arXiv:2305.01939.

[3] Li, M. and Zhang, Q., 2023. Defining and Quantifying AND-OR Interactions for Faithful and Concise Explanation of DNNs. arXiv preprint arXiv:2304.13312.

[4] Does a Neural Network Really Encode Symbolic Concepts? Mingjie Li, Quanshi Zhang Proceedings of the 40th International Conference on Machine Learning, PMLR 202:20452-20469, 2023.

This paper studies the interpretability of concepts of a DNN through the lens of AND/OR interactions. In this paper the author show that the  Harsanyi dividends studied in previous works suffers from some flaws, such as:
1) Non-uniqueness of interactions: the same boolean formula can decomposed into different AND/OR interactions. This is illustrated on a toy example.
2) In [3] the authors propose to enforce sparsity with LASSO. Unfortunately, sparse patterns are also unstable. This is verified experimentally on Cifar-10.

They propose a regularized objective to extract "robust interactions" from a classifier.

**Strengths:**

The article points out issues in previous works of literature.

The experiments cover vision datasets (e.g CIFAR) or NLP datasets

**Weaknesses:**

### Clarity

I struggled to understand the key takeaways from the paper. I am not sure what the message is here.

In the introduction, the properties like "*(1) Universal matching*" are given without proof nor references: I am not sure where in the related work I can find the proof of this fact, and it is not stated like a theorem. For "*(2) Sparsity.*" the paper you cite  Ren et al. (2023d)  does not seem peer-reviewed.

Fig 2. is unclear, the legend is insufficient.

### Link to existing literature

The non-unicity of the AND/OR decomposition is a consequence of the non-unicity of booleans circuits/formula to represent a binary function [%] But the link with this literature is not discussed.
[%] Crama, Y. and Hammer, P.L., 2011. Boolean functions: Theory, algorithms, and applications. Cambridge University Press.

A part of the article is a critique addressed toward the work of Li & Zhang (2023a):  "Sparsity does not mean stability." But to the best of my knowledge, the work of Li & Zhang (2023a) has not been peer-reviewed so i am not sure about the impact of criticizing their method.

The approach proposed, which relies on AND and OR relations, is looking at the joint effects / the independent effects of input variables, by relying on masking. This approach is also done in Sobol's indices, but a short literature review is lacking.

* Fel, T., Cadène, R., Chalvidal, M., Cord, M., Vigouroux, D. and Serre, T., 2021. Look at the variance! efficient black-box explanations with sobol-based sensitivity analysis. Advances in Neural Information Processing Systems, 34, pp.26005-26014.

* Van Stein, B., Raponi, E., Sadeghi, Z., Bouman, N., Van Ham, R.C. and Bäck, T., 2022. A comparison of global sensitivity analysis methods for explainable AI with an application in genomic prediction. IEEE Access, 10, pp.103364-103381.

> In order to extract on-manifold interactions, we decompose the network output on each perturbed [...] just like in (Li & Zhang, 2023a),

Once again (Li & Zhang, 2023a) is not peer-reviewed and I am not convinced that their method makes sense.

### Claims

> Considering that the OR interactions can also be considered as
AND interactions from another perspective (please see Appendix A for the proof), we can roughly
consider that the OR interactions in a well-trained DNN also tend to be sparse

The hypotheses behind this results are given as a footnote in p1, which make the paper hard to read. The proof in Appendix A does not check the compatibility with the original hypothesis of Ren et al (2023d)

I have the same issue with:

> In particular, on-manifold interactions mean that different perturbed inputs have similar sets of
interactions, considering it has been proven that each input only has a few salient interactions.
Noises on the input may mainly change the effect values of the shared interactions, rather than
generate a fully new set of interactions.

Is this a hypothesis of the authors, or a conclusion of the paper? How can we check the result?

Same remark for: "The K different perturbed inputs are supposed to have similar decompositions of output scores". I believe an experiment is needed.

### Other references

In the introduction, it is written:

 > Within the realm of XAI, one of the ultimate questions is whether the inference logic used by a well-trained
deep neural network (DNN) can be faithfully explained by a set of symbolic concepts or primi-
tives (Zhou et al., 2016; Wu et al., 2018; Kim et al., 2018). This question involves both mathematics
and cognitive science. Up to now, people have not reached a consensus on the definition of concepts
in DNNs, due to the inherent challenge of formulating concepts in human cognition.

You can add this recent work to the survey:

* A Holistic Approach to Unifying Automatic Concept Extraction and Concept Importance Estimation
T Fel, V Boutin, M Moayeri, R Cadène, L Bethune, M Chalvidal, T Serre, NeurIPS 2023, Advances in Neural Information Processing Systems

**Questions:**

### Q1

What is the message of the paper?

### Q2

What is a "OOD interaction" ? What is the experiment in table 2 measuring?

What is a  "On-manifold interactions" ? What is the message of figure 5 ?

### Q3

What is the difference with Sobol' indices from sensitivity analysis ?

### Q4

> Specifically, let us suppose the network output  contains a tiny Gaussian noise

Most DNN are deterministic at test time, is it realistic to assume a non deterministic behavior?